# GIS Mapping and Breast Cancer Health Care Access Gaps for African American Women

**DOI:** 10.3390/ijerph20085455

**Published:** 2023-04-10

**Authors:** Shelley White-Means, Adole Muruako

**Affiliations:** 1College of Graduate Health Sciences, University of Tennessee Health Science Center, Memphis, TN 38163, USA; 2Department of Sport Psychology, University of Mississippi, University, MS 38677, USA

**Keywords:** breast cancer disparities, mammography, racial residential segregation, income disparities, GIS analysis, African American women, structural racism

## Abstract

Black women face an unequal opportunity to survive breast cancer compared with White women. One would expect that US metropolitan areas with high percentages of Black people should report similar racial disparities in breast health. Yet, this is not the case. To provide insights about breast cancer disparities in cities with above-average and below-average racial disparities, we use GIS analysis. We depict racial composition and income categories on the same map with mammography facility locations to distinguish unique patterns of mammography access, a critical resource for breast cancer care. Looking more closely at low health disparities cities, a general and consistent pattern arises. Both White and Black people are concentrated in middle-income neighborhoods. Further, MQSA-certified facilities are not clustered in affluent areas but tend to be centrally located in the middle of the city or highly dispersed across the city, regardless of income. Our findings are consistent with the hypothesis that metropolitan areas that have a preponderance of racially segregated low-income Black households—a characteristic of neighborhoods that have experienced a history of racism and disinvestment—are more likely to experience disparities in access to primary breast care than middle-income Black, middle-income White, or high-income White neighborhoods.

## 1. Introduction

Breast cancer is the most common cancer diagnosed in women, and the second most common cause of death in women in the world [1]. Statistically, Black women face an unequal opportunity to survive breast cancer when compared with White women, although their incidence rates of breast cancer are lower than for White women (124.0 vs. 130.5, respectively) [2]. Black women are 1.41 times more likely to die [3]. There has been proven to be a genetic component to this disparity—Black women are more likely to experience triple-negative breast cancer lacking estrogen receptor, progesterone receptor, and HER2, a much more aggressive type of breast cancer that is associated with relatively higher mortality rates [4,5,6,7]. However, biology is but one explanation for these breast mortality disparities.

Another potential factor to consider and explore is the effect of geographical location on observed breast cancer survival disparities, which is the research context of this study. Specifically, Whitman, Orsi, and Hurlbert [8] suggest a geographical (place-based) dimension to racial disparities in breast cancer mortality [8]. They reported that the mortality rate ratio of deaths for Black people (44.6) divided by deaths for White people (21.3) in Memphis, Tennessee during 2005–2009 was 2.09, the highest rate among the 25 largest cities in the US at the time [8]. In subsequent studies reflecting mortality rate ratios for 2005–2009 and 2010–2014, this research team continued to find Memphis ranked at least among the top seven, with racial mortality rate ratios exceeding the national rate of 1.43 [9,10].

The Susan G. Komen Foundation [11] also reflected on the geographic dimension of Black/White disparities in breast cancer mortality [11]. Their approach to observing the geographic dimension was slightly different. They stated that while it is informative to compare cities of similar size, it is important to note that the processes of change, growth, and activity may vary depending on the racial composition of the city, which is associated with resource distribution, culture, income, and other social determinants of health (SDoH). Komen focused on eliminating mortality disparities for African American women because their mortality rate ratios were highest among all population subgroups (not only greater than Whites, but twice that of Hispanics and Asian/Pacific Islanders) [12]. Komen thought it imperative that a geographic emphasis included an apples-to-apples comparison, contrasting mortality outcomes in large metropolitan cities that also had large densities of African-Americans. Death rates, death rate ratios (ratio of death rates in Black/African American people to that of White people), late-stage diagnosis rates, and late-stage diagnosis rate ratios (ratio of late-stage rate in Black/African American people to that in White people) were examined in the 17 largest metropolitan areas that also meet the criterion of having large numbers of African Americans (Susan G. Komen African-American Health Equity Initiative, 2016). The cities were then ranked according to the burden of breast cancer disparities, with the highest burdened area ranked number 1 and the lowest burdened area ranked number 15; some cities tied for the same rank.

One would expect that US metropolitan areas with high percentages of Black residents would report similar racial disparities in breast health outcomes. Yet, this is not the case. Susan G. Komen (2016) reported that breast cancer mortality rates and mortality rate ratios were highest that year in Memphis, TN (ranked #1), Houston, TX, and Washington, DC, while substantially lower and below national rates in New York, NY, Oakland, CA, and Charlotte, NC [11]. Thus, racial density (a high percentage of Black people in a similarly sized geographical area) in and of itself does not fully mirror or account for geographic disparities in breast cancer mortality.

Our research informs the geographic emphasis on racial disparities in breast health outcomes by including a third dimension, i.e., geographical access to healthcare in marginalized communities that evolved, in part, due to a history of racism. The literature reported in the conceptual framework section guides this third strategy.

Our hypothesis is that metropolitan areas that have a preponderance of racially segregated low-income Black households, a characteristic of neighborhoods that have experienced a history of racism and disinvestment, are more likely to experience disparities in access to primary breast care than middle-income Black, middle-income White, or high-income White neighborhoods. The focus of our paper is to use GIS mapping of the overlap of racial segregation and poverty to see if this provides a better understanding of why some large metropolitan areas with high percentages of Black residents have limited access to MQSA-certified mammography (and thus limitations in efforts to achieve good breast health outcomes), while Blacks who live in other similarly sized and racially composed cities have access to certified mammography facilities and the opportunity to maximize their breast health. A focus on racially segregated low-income neighborhoods requires a focus on the overlap of racial composition and income in a neighborhood.

In this manuscript, our outcome measure is not a direct measure of breast health outcomes in metropolitan cities. For the purpose of analysis, we compare neighborhoods by the relative presence or absence of a resource that produces good breast health outcomes, namely an FDA Mammography Quality Standards Act (MQSA)-certified mammography screening facility. The viability of measuring breast health outcomes by the presence or lack of a mammography facility is explained in our variable measures section below.

### Conceptual Framework for a Third Geographic Emphasis

To add a framework for thinking about the observed geographic disparities in breast cancer mortality in cities with high percentages of Black residents, we turn to the SDoH literature [13] and the related literature on structural racism that is evidenced by racially segregated Black neighborhoods with concentrated poverty [14]. In 1977, Uri Bronfenbrenner presented a socioecological model in which health outcomes were determined by six levels of factors (organ/genetic, individual, interpersonal, institutional, community, and population). According to this framework, racial differences (or inequities) in the characteristics at any of these six levels might account for racial disparities (or inequities) in health outcomes [13]. When Komen [11] noted that cities’ resource distribution, culture, income, and other social determinants of health are related to their racial composition, they were essentially noting that cities experience differences in their community and population characteristics according to the racial makeup of the cities [11]. Thus, according to Bronfenbrenner’s model, we would expect there to be differences in health outcomes according to the racial makeup of the city, as well as differences in community characteristics that vary by race.

Urban anthropologists and sociologists report that racial inequities in community- and population-level characteristics do not always occur by happenstance but evolve via deliberate practices and policies fueled by bias and practices of structural racism—including redlining, redistricting, discriminatory lending, devaluation of assets, and scanty community investments—to maintain the status quo of racial inequality in community resources and wealth. The US Civil Rights movements of the 1950s and 1960s sought to disband separate and unequal practices in education, housing, and employment, as well as in other critical areas of basic needs. Nonetheless, “[g]enerations of economic and demographic shifts-facilitated by public policy- have produced a hyper-segregated metropolitan landscape, enabling predatory lending structures in and devaluation of minority neighborhoods…. The upshot of this is that today, the vast majority of white people live in suburbs-and white people and people of color do not live in the same suburbs…. By enormously expanding the footprint of American metropolitan settlement, suburbanization created a patchwork of communities built by design to be spatially sorted by income.” [15].

In the health equity literature, the term structural racism is used to describe metropolitan areas that are characterized by the circumstances described herein. A more formal definition of structural racism is “the totality of ways in which societies foster [racial] discrimination, via mutually reinforcing [inequitable] systems…(e.g., in housing, education, employment, earnings, benefits, credit, media, health care, criminal justice, etc.) that in turn reinforce discriminatory beliefs, values, and distribution of resources”, reflected in history, culture, and interconnected institutions [16]. Structural racism may present itself through (be visualized by) the vehicle of racial residential segregation, i.e., the physical segregation of a population by racial identity, with high concentrations of impoverished people of color within a limited geographic space. The residents of these segregated, low-income, Black neighborhoods face barriers to accessing social and economic determinants of health, including quality education, economic earnings opportunities, food security, and healthcare services [17].

A recent Urban Institute study demonstrates how this process works [18]. Reporting on capital investment in Baltimore, one of the most racially segregated cities in the United States, investments in neighborhoods with over 85 percent Black residents were three times lower than in neighborhoods with less than 50 percent Black residents. Further, high-poverty neighborhoods received two times fewer investments than non-poverty neighborhoods. The researchers noted that once a neighborhood becomes a low-investment neighborhood, it is difficult to secure investments in the future. When disinvestments occur in neighborhoods, health-promoting resources, high-quality medical facilities, and providers are more limited [19], leading to poor health outcomes [14,20,21].

There is evidence of a history of structural racism in Memphis and many other large metropolitan areas with high densities of Black persons. According to the 1935 Home Owner’s Loan Corporation (HOLC) redlining map provided by the University of Richmond’s Digital Scholarship Lab (2022), 39% of Memphis neighborhoods were rated as “hazardous” (marked in red or redlined) and 30% were rated as declining or moderately risky (yellow lined) [22]. Most of the residents of these redlined neighborhoods, even now, are Black people. Wealth accumulation (based on value of housing stock) for Black people in Memphis is lower in magnitude and has grown at a slower rate. During the years 1980, 1990, 2000, and 2010, the value of Memphis homes in which White people lived far exceeded that of those in which Black people resided, by $20,002, $37,021, $67,323, and $103,259, respectively [23]. This relatively lower wealth accumulation is associated with a poverty rate of 26.1% among Black people and 9.3% among White people [24]. It appears that structural racism that resulted from disinvestment in poor Black neighborhoods may remain a critical factor in understanding Memphis’ high racial disparity in breast cancer outcomes.

In addition to this, neighborhood poverty as a psychosocial factor has been found to be a significant determinant and predictor of health. Black/African American women across studies have among the highest rates of preventive care use and contradictorily have the largest health disparities [25]. Indeed, Black/African American women are more likely to be diagnosed with late-stage diseases, and have higher rates of morbidity and mortality than White women [26]. In regression models stratified by race where it was possible to explore within-race differences in preventive care, the study noted that preventive care was less likely for African-American women whose incomes were lower than $50,000 and who did not have a primary care provider, i.e., two characteristics most likely experienced by persons living in low-income racially segregated neighborhoods [25]. This point further compels study of potential place-based explanations.

Given a focus on racially segregated residential neighborhoods, we must acknowledge and account for the nature of the overlap of racial composition and income. That is, Black residents live in a variety of neighborhoods, not only those with a high percentage of Black residents. Racially segregated neighborhoods are also more likely to be low-income neighborhoods with high percentages of Black people residing in them. Yet, if middle-class Black people are unable to use their higher economic status to move into more resource-abundant residential neighborhoods, they may also reside in these neighborhoods [17].

Racial groups and income levels are interrelated, and both racial group and income level may influence the role of the geographic location of facilities and access to care. Indeed, some researchers have even suggested that racial group is simply a proxy for income [27]. This high correlation between racial group and income is so strongly acknowledged that it has been used in university admissions programs where income is used as a proxy for race-based admissions [28]. To account for this potentially confounding effect of income and to test the hypothesis that racially segregated impoverished neighborhoods drive breast health care disparities, we modified the analysis that we present in this paper. We explore whether the racial and income composition of a neighborhood influences access to MQSA-certified facilities in cities with high and low levels of breast health disparities.

## 2. Methods

Breast Health Outcome Measure—Presence of an MQSA-Certified Mammography Facility Instead of using a measure of breast cancer mortality as our breast health outcome measure, we use a measure indicating whether an MQSA-certified mammography facility is in a geographic area. The literature reports that early detection of disease is critical for reducing breast cancer mortality. While detection of first-stage breast cancer is associated with a 1–5% probability of mortality, its detection at stage four is accompanied by 75–80% mortality, with a median survival time of three years [29]. The preferred method of detection is mammography screening, using the resources of an FDA Mammography Quality Standards Act (MQSA) facility. The MQSA set the minimum quality standards for personnel, equipment, and record-keeping, while a facility with a higher quality rating can receive American College of Radiologist’s (ACR) Center of Excellence certification or, even higher yet, certification from the American College of Surgeons’ National Accreditation Program on Breast Centers (NAPBC). As amended in 1998, the MQSA requires that facilities have systems for communicating mammography results and for transferring the original mammograms at the patient’s request [30].

Many researchers have examined the relationship between geographical access to mammography and breast cancer screening and stage of diagnosis, providing a foundation for the use of this measure as an indicator of potential breast health outcomes. Kim, Chikwudozie, & Calhoun [31] found that the distance from one’s home to a screening facility presents a statistically significant barrier to accessing routine screening and timely follow-up of abnormal changes. The distance a woman traveled to a clinic is a significant predictor for having an abnormal mammogram [31]. Interestingly, the sample in this study was drawn from clinics that strategically target low-income, uninsured populations, and found that White women who fit this description had the least access and were more likely to have an abnormal screening result [31]. Williams and Thompson [32] found that the stage of breast cancer diagnosis is associated with racial group, age, and census tract area poverty level and that African-American women are diagnosed with a higher proportion of late-stage breast cancer than White women [32]. In a review of the literature by Kahn-Gates, et al. [33], the association between geographical access to mammography and breast health outcomes depended on the measure of access used and was either seen as being related to the increased use of screening and decreased diagnosis at a later stage or as there being no association with screening and diagnosis [33].

To provide preliminary insights about the potential association between the geographic location of mammography facilities and racial disparities in breast outcomes, we used Geographic Information System (GIS) techniques to map the location of MQSA-certified mammography centers in Memphis/Shelby County and the racial distribution of the population. Geographic Information Systems mapping handles geographic, spatial, or geospatial data for geographic research and is based on systems designed to store, retrieve, analyze, and display data that are represented spatially and geographically [34]. GIS mapping holds an increasingly critical role in documenting access to health care [35]. However, at this stage, the literature equivocates about whether GIS-based measures of access to mammography best predicts the risk of late-stage breast cancer [36,37,38].

Memphis/Shelby County contains 24 MQSA-certified facilities, which are noted in red according to their zip code location on the map in Appendix A, Figure A1, while facilities with higher certifications are noted in blue and green. One facility in this region has ACR certification and one has NAPBC certification. The percentage of the population who is African-American is also color-coded by zip code in Figure A1: the darker the shade of green used, the larger the percent population of African-American people who live in the zip code. The results show that MQSA-certified facilities are less likely to be available in zip codes where 50% or more of the residents are African-American, which suggests a disparity in access to quality mammography care in these areas.

The cities used in this research are among the 17 studied in the Komen report [11], with Komen ranking the top 10 cities that had the worst mortality and late-stage diagnosis outcomes as high disparities cities; the remaining 7 cities were classified as low disparities cities. We include a subset of the high and low disparities cities in this study: Atlanta, GA, USA; Dallas-Fort Worth-Arlington Metropolitan area, TX, USA; Los Angeles, CA, USA; Memphis, TN; St. Louis, MO, USA; and Washington, DC, USA, Baltimore, MD, USA; Charlotte, NC, USA; Cleveland, OH, USA; and Oakland, CA, USA.

### ArcGIS

ArcMap 10.1 and TIGER/Line^®^ shapefiles from the United States Census Bureau were used to create maps using the ArcMap application in ArcGIS. Data used to populate the maps were generated from the American Factfinder website, also powered by the United States Census Bureau [39,40,41,42]. The coordinate planes utilized within ArcMap were NAD_1983_2011_StatePlane projection systems for each state. Four layers representing the feature classes for analysis were used: the Zip Code Tabulation Area (ZCTA) layer, the geocoded mammography clinic locations layer, the racial demographics layer, and the aggregated income by ZCTA layer. Continuous surface depictions such as elevation, aerial photography, and satellite imagery were not utilized in the mapping analysis.

All data were aggregated to ZIP code tabulation area (ZCTA) level, a geographical unit developed by the US Census to approximate USPS ZIP code areas, with stricter geometry than traditional ZIP code areas. The 5-digit ZIP code tabulation area shapefiles for representative cities were represented as a polygon data layer and were obtained from 2010 TIGER/Line census data. Each 2010 Census tabulation block that contains addresses is assigned to a single ZCTA, usually to the ZCTA that reflects the most frequently occurring ZIP Code for the addresses within that tabulation block. The TIGER/Line Files are shapefiles and related database files (.dbf) that are an extract of selected geographic and cartographic information from the US Census Bureau’s Master Address File/Topologically Integrated Geographic Encoding and Referencing (MAF/TIGER) Database (MTDB). This polygon layer is a subset of the Census TIGER ZCTA file, re-projected to local state plane and modified by subtracting a subset of the Census TIGER water layer to create land-based boundaries.

ZCTA boundaries that define the areas of the cities of interest were identified by a web search of ZIP codes contained within each city. Each search was cross-checked to verify that overlapping ZCTAs were identified for each search within each city. Address Range Feature shapefiles (ADDREFEAT) within selected cities were also obtained from 2010 TIGER/Line census data to geocode mammography facility address locations. Mammography clinic locations were obtained from the United States Food and Drug Administration (FDA) Mammography Facility database. Facility locations within selected cities were searched using the database tool, their addresses were entered into Excel, and were uploaded within ArcMap as an Excel database file. The Geocode Addresses function was used to convert the table of obtained addresses into point features on each map, thus creating a layer.

Information on several sociocontextual factors collected from the US Census Bureau by ZCTA were included in this analysis. This information, provided by American FactFinder, was used to create choropleth maps depicting racial composition and aggregated average income by ZCTA by selected city. Cities included in this analysis were dichotomized into *high disparities* and *low disparities* cities. High disparities cities included in this analysis were Atlanta, GA; Dallas-Fort Worth-Arlington Metropolitan area, TX, USA; Los Angeles, CA, USA; Memphis, TN, USA; St. Louis, MO, USA; and Washington, DC, USA. Low disparities cities were Baltimore, MD, USA; Charlotte, NC, USA; Cleveland, OH, USA; and Oakland, CA, USA. Socioeconomic and racial composition data for each city by ZCTA were obtained from American Factfinder. Persons who did not answer either “Black or African American alone or in combination” or “White alone or in combination” on the 2010 census were excluded from the analysis. Specific data used were 2011–2015 American Community Survey 5-year Estimates, Total Population (Table B01003) [41]; 2011–2015 American Community Survey 5-year Estimates, Black or African American Alone Or in Combination with One or More Other Races (Table B02009) [39]; 2011–2015 American Community Survey 5-year Estimates, White Alone Or in Combination with One or More Other Races (Table B02008) [42]; and 2011–2015 American Community Survey 5-year Estimates and Income in the Past 12 Months (Table S1901, in 2015 Inflation-Adjusted Dollars) [40] tables. Generated tables were populated with population count information or average income information, by ZCTA. Tables obtained from American Factfinder were converted to an Excel format and joined with the spatial ZCTA shapefile layer to produce choropleth results. Each created map depicted a comparison between the “White” and “Black” population counts by income within each city. Each map utilized a visual comparison to gauge differences that could be ascertained regarding mammography facility locations and the composition of race and income levels in the neighborhoods that surrounded the corresponding locations. Black population percentages were represented with closed black circles, with a higher percentage of the Black population in each ZCTA being represented with larger closed circles. White population percentages were represented using open black circles, with larger open circles representing a higher percentage of the White population within the ZCTA. Typical groupings used for these percentages were 0–25%, 26–50%, 51–75%, and 76% as the maximum demonstrated percentage. Deviations in this categorization were seen in the Oakland, CA map, where the maximum Black population in a given ZCTA was 55%. Income level was represented by ZCTA in a choropleth format, with lower aggregate incomes shown in a lighter color and higher incomes in darker hues. Categorical income breakdowns were represented by the following groupings: $0–$35,000, $35,001–$50,000, $50,001–$75,000, $75,001–$100,000, and $100,000+. Geocoded mammography facility locations were represented as pink dots on the maps. In this analysis, multiple mammography facilities can be housed in separate building suites with the same address, each representing a unique business entity. Therefore, the number of identified mammography facilities within selected cities may not correspond to the number of points on the map. All analysis results were visually interpreted. No statistical analyses were performed in this GIS analysis.

## 3. Results

### 3.1. High Disparity Cities

Atlanta, GA, USA (Appendix A, Figure A2). A total of 40 ZCTAs with 388,009 Black and White residents were included in the analysis. The analysis area contained 25 MQSA-certified mammography facility locations. *Spatial Analyses:* The analysis of the Atlanta, GA spatial area was limited to Fulton and DeKalb counties, with a primary focus on the city center of Atlanta. ZCTAs with more limited access to mammography facilities were observed to have a racial composition that was 50% or more Black, located in the lower half of Fulton County, GA, as is visually evident in the map. The majority of the available mammography facilities in the area (22 facilities) were found in ZCTAs that had 51% or more of residents identified as White and that had incomes of at least $75,000 or more.

Dallas-Fort Worth-Arlington, TX, USA (Appendix A, Figure A3). A total of 96 ZCTAs with 1,783,806 Black and White residents were included in this analysis. The analysis area of Dallas-Fort Worth-Arlington, TX contained 110 available mammography facilities. *Spatial Analyses:* The majority of the ZCTAs in the Dallas-Fort Worth-Arlington Metropolitan area had a racial composition that was less than 25% Black and at least 51% White. Most ZCTAs within the analysis area with higher Black populations had aggregated incomes of $50,000 or less; the southern half of the map area contained greater than 51% of the Black population. The majority of the available mammography facilities in the area (23 of the 28 facilities depicted) appeared to be clustered in the center of the metropolitan area, in ZCTAs with an aggregated income of $75,000 or more, while five points that represented mammography facility locations were located in ZCTAs with incomes of $50,000 or less.

Los Angeles, CA, USA (Appendix A, Figure A4). A total of 66 ZCTAs with 2,253,276 identified Black and White residents were included in this analysis. The analysis area of Los Angeles contained 62 available mammography facilities. *Spatial Analyses:* Los Angeles as a whole contained a large White population, with the majority of ZCTAs containing 51% or more White population. One cluster of ZCTAs containing a Black population that was 61% or more was located in the east-central portion of the analysis area. Mammography clinics were generally positioned in the majority of ZCTAs in the Los Angeles area. Notably, there were no mammography facility locations in the east-central portion of the analysis area where the largest clustering of the Black population was located.

Memphis, TN, USA (Appendix A, Figure A5). A total of 28 ZCTAs containing 599,807 Black and White residents were included in this analysis. The analysis area of Memphis contained 25 available mammography facilities. *Spatial analysis*: The majority of ZCTAs in Memphis, TN contained at least 51% Black population, but those with the highest aggregated incomes were composed of a population that was less than 50% Black. The majority of ZCTAs in the city with 51% or more Black population had aggregate incomes of $30,000 or less. Available mammography facilities in the Memphis, TN area appeared to have more space availability across ZCTAs with aggregate incomes of $100,000 or more. ZCTAs with aggregate incomes of $75,000 or less appeared to have less spacing between available facilities.

St. Louis, MO, USA (Appendix A, Figure A6). A total of 45 ZCTAs containing 297,427 Black and White residents were included in this analysis. The analysis area of St. Louis contained 25 available mammography facilities. *Spatial analysis:* ZCTAs in the St. Louis area with lower aggregated incomes ($30,000 or less) were localized to the east-central portion of the map, with higher aggregated incomes ($50,000 or more) located on the outside portions of the map area. ZCTAs with aggregated incomes of $30,000 or less also had populations that comprised 51% or more Black individuals. Most available mammography facilities (24 locations) were available in ZCTAs with $50,000 income or more, with populations that were 51% or more white. One mammography facility appeared in a ZCTA with an aggregate income of $30,000 or less; this ZCTA also had a Black population of between 51% and 75%. Four ZCTAs with populations that were more than 51% Black contained mammography locations in the area, compared with mammography facility locations in eighteen ZCTAs with populations that were 51% or more White.

Washington, DC, USA (Appendix A, Figure A7). A total of 28 ZCTAs containing 536,596 Black and White residents were included in this analysis. The analysis area of Washington, D.C. contained 14 available mammography locations. *Spatial analysis:* Upon visual inspection, only two mammography clinics were located in ZCTAs with incomes less than $30,000, while the remaining 12 mammography locations were located in ZCTAs where the median income was $75,000 or more. ZCTAs with Black populations of less than 25% contained 10 of the 14 available mammography facilities.

### 3.2. Low Disparity Cities

Baltimore, MD, USA (Appendix A, Figure A8). A total of 39 ZCTAs with 579,611 Black and White residents were included in this analysis. This analysis area contained 24 available mammography facilities. *Spatial analysis:* Visually, mammography facility locations appeared to be distributed throughout the north, south, east, and west quadrants of the analysis area and multiple mammography facility locations were available in all choropleth income categories. Additionally, only one ZCTA in the city had an aggregate income of $34,000 or less. Multiple mammography facilities appeared to be located in ZCTAs with an aggregate income of $34,001–$50,000 and with at least 51% Black population.

Charlotte, NC, USA (Appendix A, Figure A9). A total of 24 ZCTAs with 621,625 Black and White residents were included in the analysis. There were 19 available mammography locations. *Spatial analysis:* Sixteen of the 19 available mammography locations were located in ZCTAs with a median income of $75,000 or more, the majority of which were also at least 51% White. The other three mammography centers were located in a ZCTA with a median income of $50,001–$75,000 and a population that is at least 41% Black. The lowest-income ZCTA had a majority Black population and was directly proximal to high-income ZCTAs of $100,000 or more, where the majority of the available mammography facilities in the area were housed.

Cleveland, OH, USA (Appendix A, Figure A10). A total of 47 ZCTAs with 359,601 Black and White residents were included in the analysis. There were 38 available mammography locations. *Spatial analysis:* Visually, mammography clinics appeared to be heavily distributed across the entire map area of Cleveland, with the exception being that there were no mammography clinics located in the southernmost ZCTAs, which had incomes of $100,000 or more. Additionally, there appeared to be less overlap in terms of multiple mammography facilities being housed within one location; most mammography facilities in the Cleveland area were in standalone locations. In the northeastern portion of the Cleveland analysis area, there was an area that appeared to house a Black population of 51% or more, while in the majority of the ZCTAs in the analysis area, the population corresponded to at least 51% or more White. Mammography facilities in the Cleveland area appeared to be located across all income and racial group levels.

Oakland, CA, USA Appendix A, Figure A11). A total of 15 ZCTAs with 244,396 Black and White residents were included in the analysis. There was a total of eight available MQSA-certified mammography clinics in the analysis area. *Spatial analysis:* Of the eight mammography clinics available in the Oakland, CA area, seven facilities were located in a single ZCTA (90609), where the median income level is $75,000 or more. One mammography clinic was located in an area where the median income was between $45,001 and $75,000. The population by ZCTA was equally distributed between persons identifying as Black and white within each ZCTA boundary, except for the ZCTAs in the city with the highest median income.

## 4. Discussion

By depicting both racial group and income on the same map with mammography facility locations, we can distinguish between the relative roles of racial group versus income in access to reliable mammography. Mammography centers on the maps generally appeared to be located in higher-income areas that were more likely to house a population that contained 50 percent or more White people. Neighborhoods with a high percentage of Black people were also more likely to have higher concentrations of poor residents. Highly segregated White neighborhoods were more likely to have higher-income residents. Visually, we notice large clusters of MQSA facilities located in high-income White residential areas.

Looking more closely at cities with low health disparities, we observed a general and consistent visual pattern. Both White people and Black people were concentrated in middle-income neighborhoods. In the city of Oakland, CA, USA, a high-income area with incomes of $100,000–115,000, there was a concentration of Black people. In these low disparity cities, MQSA-certified facilities did not appear to be clustered in affluent areas but rather tended to be centrally located in the middle of the city or highly dispersed across the city, regardless of income. Thus, access to facilities may be more readily available to neighborhoods of all racial and income compositions. This greater equality of access was consistent with lower racial disparities in breast health outcomes. Unlike cities where structural racism and disinvestment of resources exist for Blacks who live in low-income areas, this centralization of resources provides a healthcare policy solution for eliminating inequitable access to services by mitigating distance from home to screening services [31]. The challenge is the process that might be involved in getting MQSA-certified facilities to relocate, thus enabling earlier screening, diagnosis, and treatment; reduced premature loss of life; lower treatment costs for healthcare services; and lower Medicaid expenditures.

The GIS maps for high disparities cities indicated the presence of varying and unique patterns for the location of MQSA facilities relative to population racial group and income distribution. In Atlanta, there are no low-income and predominately White zip codes, and middle-income White and Black neighborhoods were located on different sides of the city, geographically separated by high-income White neighborhoods. Mammography facilities were concentrated in high- and middle-income White neighborhoods. In Dallas, more White people lived in wealthy and middle-income neighborhoods, and the mammography facilities were concentrated in those neighborhoods. In Memphis, there was only one predominantly Black middle-income neighborhood, and the mammography facilities were located in higher-income, predominantly White neighborhoods. In St. Louis, most predominantly Black neighborhoods were low-income areas, with mammography facilities clustered in predominantly White high- and middle-income neighborhoods. Washington, DC, USA was similar to St. Louis in its racial group/income distribution of the population, yet its mammography facilities were centrally located, but only among high- and middle-income White and Black neighborhoods.

Los Angeles, a high disparity city, was an anomaly among the cities we studied in providing a depiction of the role of racial group vs. income in the location of mammography facilities. Only one area in the city had a large Black population, but regardless of neighborhood wealth or poverty, mammography clinics were located everywhere in the city except in that cluster.

Structural racism in this study was exhibited through visualization of resources placed in areas of high economic means and racialized residential segregation. This problem is evidenced in suburbanization. Extensive research findings consistently show that suburbanization continues to contribute to the separate and unequal distribution of resources. Community context and segregation dramatically inform outcomes and contribute to continued barriers in healthcare and the propagation of disparities.

Strengths of the present study include a clear visualization of MQSA facilities in juxtaposition to demographic composition, providing an opportunity to directly view the relationship between geographic locations and to compare with documented disparities. The ability to create multiple perspectives through visualization enhances perceptive abilities and furthers the disparities work initiated by the Komen Foundation. This study clearly exhibits an inequitable distribution of resources and demonstrates this in stark form. While we do not attempt to explain all the factors that lead to the disparities in access to mammography facilities, we do focus on a less-explored factor: providing a unique and close examination of the impact of geographic segregation of low-income Black women on their access to breast cancer care, a factor that also may influence chances for breast cancer survival. This research contributes to the literature by linking the literature on structural racism with the literature on healthcare disparities among breast cancer patients. Structural racism is a process that leads to the creation of separate but unequal distribution of resources. We provide a clear visual picture of the implications of separate but unequal processes for access to breast cancer care; inequities occur in the location of a critical resource for women at risk of breast cancer, i.e., mammography facilities. These facilities are more likely to be found in middle-income Black, middle-income White, or high-income White neighborhoods, while separate low-income Black neighborhoods lack access.

## 5. Limitations

This study, as it represents a pilot for a more intensive examination of the concepts within, has some limitations. Black and White populations were the only examined racial groups—other populations were excluded to demonstrate the concept most simply, especially considering that other racial compositions are much smaller and population clusters may be negligible for GIS mapping. White populations, since they comprise the majority population, were exhibited as a counterfactor and to highlight key differences. However, this is not to say that other groups are not potentially affected by structural racism.

The main analysis of this pilot study was to visually capture differences in location of MQSA-certified breast cancer facilities and the demographic makeup. We found that some of these visual differences were stark. However, further research that incorporates statistical analysis is warranted. Additionally, the analysis could be extended to consider other types of health-producing resources, such as hospitals, breast cancer clinics/centers, and general practitioners and oncologists.

## 6. Conclusions

Analyses for this paper were based on visual observation. Although use and experience of residents in the various ZCTAs under analysis cannot fully be known within the scope of this study, it is clear that MQSA facilities were consistently absent in lower-income areas with predominantly African-American residents, particularly in high-disparities cities. This visual analysis presents opportunities to shape the policy discourse, and further research is needed. The association between racial residential segregation and poverty is critical for breast cancer care and outcomes.

Further analysis of our hypothesis using empirical methodology that includes other causal influences is warranted. If such methodologies identify that an intersectionality of racial residential segregation and poverty was a significant determinant of mammography facility location, public policy intervention at the municipal level may be needed.

When exploring healthcare policy interventions at the state and federal levels, the general focus is on requisite changes in health insurance coverage or strategies to change the behavior of patients or providers. However, healthcare is not in a silo; it is impacted by decision-making in other sectors, including those that are not focused on producing health. As our research suggests, structural characteristics of the communities in which one lives, works, and plays also impact health and access to healthcare. It is critical to note that public policy resolutions will require explorations of the historical context in which laws, policies, and practices evolved in particular geographic areas, as well as an acknowledgment that racially disparate health outcomes is one of the costs of not assuring that all individuals experience equitable access to healthcare services.

## Data Availability

No new data were created or analyzed in this study. Data sharing is not applicable to this article.

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
