# Peer review of "GIS Mapping and Breast Cancer Health Care Access Gaps for African American Women"

_ijerph, 2023, doi:10.3390/ijerph20085455_

Round 1
Reviewer 1 Report
The article proposed for my review is entitled “GIS Mapping and Breast Cancer Health Care Access Gaps for African American Women”.
This article begins with a rather lengthy introduction that outlines the breast cancer mortality risks of black women. In doing so, they indicate that black women are 1.41 times more likely to die. Based on the work of Whittman et al (2012), the authors suggest that there is a geographic dimension explaining racial disparities in breast cancer mortality. In the 50 largest cities in the US, the breast cancer mortality rate for black women appears to be on average 2 times higher than that of white women. The authors also mention that the Susan G. Komen Foundation (2016) postulates that the racial component of cities can explain disparities in breast cancer mortality between black and white women.
A study by this foundation shows that breast cancer mortality rates and mortality rate ratios are higher in Memphis, TN, Houston, TX, and Washington, DC, than in New York, NY, Oakland, CA, and Charlotte, NC. While the presentation of this result seems important, for a non-North American who is unfamiliar with the racial composition of these six cities, it is difficult to fully understand this result. The racial composition of the above cities should be clarified. In addition, the presentation of the result graph (graph 1) is quite unusual in an introduction. In general, in the introduction to a scientific article, the results of previous studies and work are presented in a very synthetic manner in one or two sentences and not in the form of graphs.
Following the presentation of this study, the authors conclude their introduction by mentioning that racial density (a high percentage of black people in a similar sized geographic area) does not fully reflect or account for geographic disparities in breast cancer mortality. In contrast, this introduction does not present a clearly presented research question or objective. Thus, a research gap and, more importantly, a research question should be accurately identified and described. Then, the methodology to answer this research question should be described. Finally, the introduction usually presents the structure and the different parts of the article. Indeed, the research hypothesis of this work is presented rather late (p. 4) and before this page 4, it is difficult to understand what the exact research objective of this article is. The research gap and the research question should be clearly stated in the introduction.
The first part of the article presents the conceptual framework for this study. This conceptual framework, as presented, primarily describes a literature related to racial residential segregation in cities (Bronfenbrenner, 1977; Komen (2016) and structural racism (Williams et al., 2019). If this literature is interesting, is there not a literature addressing health disparities across cities and specifically health disparities in cancer or breast cancer care? If so, it would also be interesting to describe it.
The authors conclude this conceptual framework section by suggesting that structural racism resulting from disinvestment in poor black neighborhoods may be a key factor in understanding the large racial disparity in breast cancer mortality. How can disinvestment in poor black neighborhoods be considered to stem entirely from structural racism? How do previous studies justify this relationship? There is not enough justification for the link between structural racism and disinvestment in black neighborhoods. While this relationship is probably defensible, it is simply asserted by the authors but never scientifically justified. This greatly weakens the relevance of its mobilization in the article. It would be more appropriate to link disparities in breast cancer mortality with territorial disparities without necessarily connecting these disparities with structural racism unless the concepts of "structural racism" and "racial residential segregation" are better defined and then mobilize data that justify structural racism or racial residential segregation in the cities studied.
The next section of the article presents the article's Research Hypothesis and Methodological Foundation. The hypothesis defined by the authors is that “metropolitan areas that have a preponderance of racially seg-regated low-income Black neighborhoods that have experienced structural racism are more likely to experience disparities in breast health outcomes than middle-income Black, middle-income white, or high-income white neighborhoods”.
While this assumption is clear enough, asserting that low-income black neighborhoods would be racially segregated and structurally racist seems to us to have little justification in the earlier parts of the article.
Also in this " Research Hypothesis and Methodological Foundation" section, the authors note that many researchers have examined the relationship between geographic access to mammography or screening and breast cancer mortality (Kim et al., 2013; Williams and Thompson, 2017). This part is interesting but should not be in the methodological section but rather in the conceptual framework section.
In this methodological section the authors also mention that they wanted to explore whether racial group membership was related to the location of mammography care. In our opinion, this could be the research question of the article and should have been presented in the introduction. However, it only appears on page 5 of the manuscript. In our opinion, the presentation of this research question comes too late.
In methodological terms, the authors go on to report using GIS techniques to map the location of MQSA-certified mammography centers in Memphis/Shelby County and the racial distribution of the population. It should be clarified what the acronym GIS stands for and what it consists of.
Figure 1 shows a map of the 24 MQSA-certified facilities in Memphis/Shelby County. This county seems to represent the field of study for this article, but subsequently other American cities are also taken as a field of study. At this point in the article, it would be appropriate to be more specific about the exhaustive list of cities that will be studied in this article.
In a section entitled "Methods", the authors indicate that they used ArcMap 10.1 software. In this section, the authors indicate precisely the origin of the data used and the ways in which they were processed using ArcMap 10.1. This "methods" section is complete and provides a good description of the authors' methodological approach.
In the results section, the authors present key maps developed from the data collected. The maps show the location of black and white populations in relation to MQSA-certified facilities in the cities of Dallas, Atlanta, Los Angeles, Memphis, St. Louis, Charlotte, Washington, Baltimore, Cleveland and Oakland. First, it is important to clarify why these 10 cities were chosen for study and not other U.S. cities. The authors indicated that they also studied New York City, but the maps and results for this city do not appear. Why ?
We can also wonder about the location of Asian or South American populations on these maps. Are they considered white populations? Do they not have particularities in terms of breast cancer mortality and geographical location in the cities studied? Are they not possibly victims of racial residential segregation or structural racism? Without answers to these questions, it is difficult to be completely convinced by the results presented in this article.
In addition, each map shows the location of black populations on one side and the location of white populations on the other. Wouldn't it be simpler to present the location of the black and white populations on the same map since they are the same geographic areas and the same MQSA facilities? This would make it clear that certified facilities are primarily located in CAZs with a majority white population, which seems to be the main finding of this geographic analysis.
The discussion section seems to confirm this main finding. The authors note that on the maps, mammography facilities are generally located in high-income areas, which are more likely to have a population that is 50% or more white.
The authors seem to want to indicate that in cities where there is relatively little socioeconomic disparity and separation between black and white populations (Oakland), MQSA-certified institutions seem to be well distributed throughout the city. This is an original and interesting line of discussion, but it should be discussed and explained further.
The authors conclude their discussion by noting that in this study, structural racism manifested itself in the visualization of health resources placed in areas of high economic means and racialized residential segregation. On the one hand, it seems inappropriate to me to indicate that structural racism is the cause of the location of MQSA facilities in cities, or at least it is not scientifically proven by this study. The analysis of the maps shows that it is economic or social disparities that are probably at the origin of the unequal geographical location of health resources. However, there is no evidence to suggest that these disparities are due to structural racism. Again, the concepts of "structural racism" and “racial residential segregation" are poorly explained in the article, making it very difficult for the reader to link socioeconomic and territorial disparities with structural racism or racial residential segregation.
On the other hand, these results do not answer the question that was more or less mentioned in the introduction, which sought to explain the differences in breast cancer mortality ratios for black and white populations. The results of this study suggest that black populations are, in cities, further away from MQSA certified facilities. This result is simply one possible explanation for the differences in mortality between these populations, but there are surely others (medical, social, economic). In my opinion, the introduction should be completely restructured to make it clear that this work will focus on the location of black and white populations in relation to facilities and health care professionals providing breast cancer screening and care. Indeed, since breast cancer care is not limited to screening in MSQA certified facilities, it may be necessary to include in the geographic mapping analysis other facilities and health care professionals involved in breast cancer care (hospitals and breast cancer clinics, general practitioners, nurses, rehabilitation facilities, etc.). This would confirm or refute the results obtained for the location of MQSA screening facilities.
In the conclusion, the authors confirm the hypothesis that neighborhood segregation by race and income is associated with inequalities in access to mammography screening. Through the results of this study, it seems to us that the explanation of unequal access to mammography screening by racial residential segregation or structural racism is not justified.
In conclusion of this article, I would say that this work, if it approaches an important and interesting topic suffers from many flaws that I synthesize below:
- An introduction that is not sufficiently synthetic and precise and does not present a clear theoretical gap and research questions.
- A literature review focused on work on racial residential segregation and not sufficiently oriented to territorial disparities in cancer care and breast cancer specifically.
- An overly imprecise definition of the concepts of "structural racism" and "racial residential segregation" that seem to be the conceptual framework of this work.
- A geographic mapping methodology that is well explained but has insufficient data on facilities and health professionals involved in breast cancer care.
- A discussion of findings that assert that disparities in breast cancer care between black and white populations in the U.S. stem from racial residential segregation and structural racism without actually proving it scientifically.
As it stands, you may be able to address the first three flaws. To answer the next two flaws, you need to mobilize new data and then process and discuss the results, which can be more complicated.
I wish you good luck for the continuation of your work and I thank you for having given me the opportunity to read your work on a very important but difficult theme.
Yours sincerely
Reviewer 2 Report
- What was the method/range for the dichotomization of high disparities and low disparities? Why did you select that classification range? This is not clear.
- It would be good to explore mitigation strategies of adjusting related resource allocation and maximizing the benefits of MQSA centers.
- Discussion of particular interventions which are effectively targeted are useful. How to integrate spatial and nonspatial factors to characterize access to medical services.
- Were there any explorations of how the other levels of factors in the socioecological model noted in the conceptual Framework affect breast health disparities? Age? Utilization? Mammogram in the last 1-2 years; screening trends? Greater stratification of breast health disparities and rate ratios? Incorporation of other health indices? Incidence-to-mortality? Stage at diagnosis? Survival outcomes?
- A greater discussion of the strengths and limitations of the study would be useful.
- There are some grammatical and formatting errors.
- The data visualization is quite excellent.
- The image resolution of Map 1 needs to be increased.
- The image resolution of Map 6 needs to be increased.
- A descriptive title needs to be added to the Chart 1 caption.
- Greater context about the barriers of access in neighborhoods of different socioeconomic status would be helpful to the reader. Further statistical analysis which analyzes influence of spatial access to other breast health indicators such as disparities at stage of diagnosis and outcomes would be really helpful.
- Excited to see what comes next.
Round 2
Reviewer 1 Report
I thank the authors for their work to improve their article
The introduction has been very sensitively modified and provides clearer information. The hypothesis of this article is clearly stated. It is to show that metropolitan areas that have a preponderance of racially segregated low-income black households are more likely to experience disparities in breast health outcomes than middle- or high-income black neighborhoods. The research objective being to use GIS mapping of the overlap of racial segregation and poverty to see if this provides a better understanding of why some large metropolitan areas with a high percentage of black population have limited access to certified mammography. The conceptual framework for this study is more developed. Structural racism is much better defined and the methodology is much more precise
Nevertheless, this article leads me to make a few additional comments and suggestions:
1. While the authors hypothesize that metropolitan neighborhoods with high black populations experience disparities in breast cancer care, the authors do not mobilize medical data related to breast cancer. They use only geographic data related to the location of patients and MQSA certified mammography. Thus, the hypothesis should be about access to primary breast care and not about disparities in breast cancer care.
2. In their literature review, the authors report that black or African American women have the highest rates of preventive care utilization (of which certified mammography) and, contradictorily, have the greatest health disparities (Veldhuis, Maki, & Molina, 2020). Therefore, in their study, can the authors explain this contradiction from the results? Indeed, the authors mention that disparities are due to delays in care or premature termination of treatment (Veldhuis, Maki, & Molina, 2020). How can disparities be explained as due to delays in care when black or African American women have the highest rates of preventive care use??? And how could location explain disparities when black or African American women (some of whom live in segregated neighborhoods) use more preventive care? You should clarify why you are studying the influence of location of preventive care (MQSA certified mammography) when black women have a higher rate of use of preventive care
3. In the methodology, we now understand that 17 cities were studied. 10 cities considered high disparity identified by Susan Komen's report and 7 cities added by the authors to the sample. Since the reader may not be familiar with Susan Komen's report, wouldn't it be simpler to present a table showing the cities studied, their populations, the share of black population, etc.? Why is figure 1, which is a map, not called map 1?
4. In the article contributions, you mention that your study measures the impact of geographic segregation of low-income black women on their chances of breast cancer survival. I don't think you fully demonstrate this because you don't cross-reference the data on the location of patients relative to the MQSA certified mammography with the survival rate of those patients. You simply measure the distance of patients from segregated metropolitan areas to certified mammography. This simply allows you to hypothesize that this distance could explain the lower survival rate of these patients but you do not scientifically demonstrate this.
5. the conclusion indicates that the approach of this study is essentially visual and allows for recommendations to be made to policy makers in particular. It would be important for the authors to also return to their theoretical contribution(s). In what way does this study contribute to the work that has specifically studied health disparities in cancer and breast cancer (Komen, 2020. 2016; Veldhuis et al. 2020)? In my opinion, the main value of this study is that it shows that MQSA certified mammography tend to be located in middle to high income areas and few in areas with a majority black population. The development of suburbanization in the USA then helps to explain the separate and unequal distribution of these resources.
In my opinion the theoretical contribution is still small and the authors should explain in what ways this result contributes to the literature on healthcare disparities of breast cancer patients or possibly to the literature on structural racism. Indeed, you mention structural racism as a conceptual framework for your study and clearly define it. But aren't you also contributing to that literature?
Thank you very for your efforts to improve your article and for your explanations
